# AIRFRANS: High Fidelity Computational Fluid Dynamics Dataset for Approximating Reynolds-Averaged Navier–Stokes Solutions

**Florent Bonnet**
Sorbonne Université, CNRS, ISIR
Extrality
Paris, France
bonnet@isir.upmc.fr

**Jocelyn Ahmed Mazari**
Extrality
Paris, France
ahmed@extrality.ai

**Paola Cinnella**
Sorbonne Université, Institut Jean Le Rond d'Alembert
Paris, France
paola.cinnella@sorbonne-universite.fr

**Patrick Gallinari**
Sorbonne Université, CNRS, ISIR
Criteo AI Lab
Paris, France
patrick.gallinari@sorbonne-universite.fr

## Abstract

Surrogate models are necessary to optimize meaningful quantities in physical dynamics as their recursive numerical resolutions are often prohibitively expensive. It is mainly the case for fluid dynamics and the resolution of Navier–Stokes equations. However, despite the fast-growing field of data-driven models for physical systems, reference datasets representing real-world phenomena are lacking. In this work, we develop AIRFRANS, a dataset for studying the two-dimensional incompressible steady-state Reynolds-Averaged Navier–Stokes equations over airfoils at a subsonic regime and for different angles of attacks. We also introduce metrics on the stress forces at the surface of geometries and visualization of boundary layers to assess the capabilities of models to accurately predict the meaningful information of the problem. Finally, we propose deep learning baselines on four machine learning tasks to study AIRFRANS under different constraints for generalization considerations: big and scarce data regime, Reynolds number, and angle of attack extrapolation.

## 1 Introduction

Numerical simulations of physical dynamics are a consequent part of scientific research as it allows us to quantitatively study natural phenomena without requiring often complex and expensive experiments. Those dynamics are mainly governed by Partial Differential Equations (PDE) and are numerically solved with the help of discretization methods such as finite differences, finite elements, or finite volumes methods. Such techniques are accurate when used over sufficiently fine meshes but are often expensive in time and resources. Thus, the optimization of meaningful quantities with respect to the parameters of the studied dynamics is, most of the time, out of scope. In particular, the numerical

36th Conference on Neural Information Processing Systems (NeurIPS 2022) Track on Datasets and Benchmarks.

resolution of Navier–Stokes equations for fluid dynamics analysis leads to computations that can last for thousands of CPU hours. Hence, the design of accurate surrogate models is at the core of engineering as they allow us to tackle the task of optimization via data-driven approaches. However, to be able to compare and validate such surrogate models we need datasets of reference and evaluation protocols. For physical systems, some efforts have already been done in this direction [45, 4] and our work is another contribution to those efforts. In [4], we developed the first version of this dataset to study Reynolds-Averaged Navier–Stokes (RANS) equations with Machine Learning (ML) models along with an appropriate evaluation protocol. In this paper, we propose an extension of this work by introducing a new high-fidelity version of the dataset. This high-fidelity version is built over finer meshes than the previous one which helps to fight numerical diffusion and allows us to recover more accurate fields and the trail of airfoils. Moreover, it allows us to accurately compute the force coefficients acting over geometries.

We focus on the classical aerodynamics task of predicting the steady-state two-dimensional fields and the force acting over airfoils in a subsonic regime. The ultimate goal of this task is to be able to find the best airfoil in terms of lift over drag ratio (see chapter 1 of [2]) in addition to the associated velocity and pressure fields. It is already a non-trivial problem in Computational Fluid Dynamics (CFD) as turbulence is involved and mesh engineering is required to find accurate force coefficients. To accelerate the resolution process, different ML frameworks can be used to build surrogate models [29, 20]. Deep Learning (DL) is among the successful candidates and has recently gained popularity for fluid simulation [58]. Moreover, the emerging field of Geometric Deep Learning (GDL) [7] models allows us to achieve learning directly on unstructured data [47] which, in this particular case, allows us to compute accurately meaningful quantities at the surface of geometries.

In this work, we present a high fidelity aerodynamics dataset of RANS solutions around airfoils. In Section 3 we present the RANS equations, the chosen design space for the airfoils generation, the meshes construction, and the dataset generation procedure. We also present the two force coefficients of interest, namely the drag and the lift coefficients. In Section 4 we introduce the different sub-tasks of the problem in addition to the evaluation protocol and the setup for our GDL baselines. In particular, the evaluation protocol contains metrics and visualizations for the force coefficient ranks and the accuracy of the surrogate models over boundary layers. We finally present, in Section 5 the results of our baselines on the main sub-task and let the remaining ones in Appendix M. All the values of the constant used in this work and the definition of dimensionless quantities are given in Appendix E.

## 2 Related Work

Although several research directions are established to come up with efficient surrogate models to tackle physics problems, from physically guided methods [15, 13, 53, 40, 6] to neural operators [37, 31, 38, 41] and Physics Informed Neural Networks (PINN) [48], the lack of standard benchmarking datasets, and common evaluation protocols impede making rigorous comparisons between the different families of methods for a given task. Benchmarking datasets and common evaluation protocols are shown to be the key components for making progress as it is observed in neighboring fields such as, for example, computer vision [32, 9] and speech recognition [1]. Though, few physics-based datasets have been proposed such as: 1D Burger's equation and 2D Darcy flow PDE [37], structural mechanics [47], incompressible fluid in vorticity form [38], reaction-diffusion, wave-equations and damped pendulum [67], heat transfer equation [68], Lorenz system [18]. More recently, few standard benchmarks datasets on complex chemical and physical systems [59, 17, 22, 21, 24] have been proposed. More interestingly, [45] suggests a framework to study a set of representative physics problems with appropriate evaluation protocols, namely a single oscillating spring, a one-dimensional linear wave equation, a Navier–Stokes flow problem, as well as a mesh of damped springs. We follow those efforts by proposing a dataset on a steady-state aerodynamics task with dynamics that can be found in realistic flight scenarios. We also focus the validation of models on meaningful parts of the dynamic instead of only regarding the mean square loss of regressed fields.

Most of the works proposed in the literature to tackle tasks represented by Navier–Stokes equations are grid-based approaches [60, 58, 43, 62, 44, 26, 63] which rely on Convolutional Neural Networks (CNN). Other architectures such as Fourier Neural Operator [38] act in the frequency domain and require a regular grid to perform a Fast Fourier Transform of the input data. Those models are not designed to directly operate on unstructured data like CFD meshes, resulting in inaccurate predictions of the physical fields at the surface of geometries. However, recent progress in learning

on unstructured data [7] has enabled learning on graphs and manifolds by designing geometrical inductive bias in DL [25, 50, 36, 30, 49]. This framework is particularly useful to achieve learning on arbitrary shapes and frees us from the constraint of data voxelization as required by CNN. One successful attempt at learning Navier–Stokes (or RANS) equations with Graph Neural Networks (GNN) can be found in [47]. Finally, let us emphasize that PINN, as defined in [48] can act on unstructured data but are not suited for surrogate modeling as they are designed to solve one and only one PDE.

## 3 Dataset Presentation

**Design-oriented dataset.** This dataset is mainly motivated by a realistic shape optimization problem. We choose a classical aerodynamics problem for this purpose: airfoil design optimization. The goal is to accurately predict force coefficients in addition to the different fields of the fluid in a subsonic flight regime with a reduced quantity of data as is often the case in practice. The design space is chosen from NASA's early works on airfoils via the 4 and 5 digits series [14] as they are easy to handle and already rich families of shapes.

We aim to resolve the air dynamic around a two-dimensional (2D) airfoil in a steady-state subsonic regime at sea level and $298.15\,\mathrm{K}$. More precisely, we study airflows at a Reynolds number between 2 and 6 million, which leads to turbulent behavior of the fluid. It corresponds to a Mach number smaller than 0.3 which allows us to assume incompressible flow behavior (see chapter 8 of [2]), and a velocity greater than $30\,\mathrm{m\,s^{-1}}$ which is a reasonable lower bound in subsonic flight conditions. Moreover, as the flow is turbulent in certain areas, we use Reynolds-Averaged-Simulations (RAS) with a sufficiently high number of cells in our meshes to accurately compute the force acting over airfoils. This method solves the RANS equations widely used in Computational Fluid Dynamics (CFD) to control the numerical complexity of the resolutions.

We rely on the Turbulence Modeling Resource (TMR) of the Langley Research Center of the National Aeronautics and Space Administration (NASA) [10, 12, 33, 61] to generate our dataset. In what follows, we present the different steps to build the dataset, and we define the relevant physical quantities of the problem.

**Reynolds-Averaged Navier–Stokes equations.** At a high Reynolds number, untidy patterns emerge in fluid flows; we call this phenomenon turbulence. In CFD, turbulence resolution is a crucial problem as it implies transient simulations on prohibitively fine meshes most of the time. Different strategies have been developed to tackle this problem, one of them being RAS. In RAS, we solve mean-field equations similar to Navier–Stokes equations but with an effective viscosity representing the diffusion added through turbulent processes. Those equations, called incompressible RANS equations, are given by:

$$\partial_i \bar{u}_i = 0 \tag{1}$$

$$\partial_j(\bar{u}_i \bar{u}_j) = -\partial_i \left( \frac{\bar{p}}{\rho} \right) + (\nu + \nu_t)\partial_{jj}^2 \bar{u}_i, \quad i \in \{1, 2\} \tag{2}$$

where $\bar{\cdot}$ denotes an ensemble-averaged quantity, $\partial_i$ the partial derivative with respect to the $i^{th}$ spatial components, $u$ the fluid velocity, $p$ an effective pressure, $\rho$ the fluid specific mass, $\nu$ the fluid kinematic viscosity, $\nu_t$ the fluid kinematic turbulent viscosity and where we used the Einstein summation convention over repeated indices. Often, in the incompressible case, the effective pressure is replaced by the reduced pressure, abusively denoted by the same symbol via the transformation $p \to p/\rho$, which allows us to write RANS equations without explicit dependence on $\rho$. From now on, we will only discuss in terms of the reduced pressure. Finally, the dynamics of the turbulent viscosity is driven by a set of supplementary equations. In this work, we choose to use the well-known $k - \omega$ SST turbulence model [42] which is well suited for aerodynamics problems. Details on the RANS equations, the definition of the different quantities, the ensemble average, and the choice of the turbulent model are given in Appendix F.

**Airfoil design space.** In the first half of the twentieth century, teams of the National Advisory Committee for Aeronautics (NACA) worked on several airfoil families. Two of them called the 4 and 5 digits series, are entirely parameterized and allow us to generate a broad spectrum of airfoils quickly. Both series define a camber line and an envelope around this camber line. An airfoil of the 4

Table 1: Sampling strategy for generating airfoils from the 4 and 5 digits series. An interval or a discrete set means that the sampling is uniform over this set. In the 4 digits case, the sampling for P has been a uniform sampling on the interval $[0, 7]$, and all the samples smaller than 1.5 have been set to 0 to get rid of geometries that have their maximum camber too close from the leading edge.

| 4-digits | | | | 5-digits | | | |
| --- | --- | --- | --- | --- | --- | --- | --- |
| M | P | XX | | L | P | Q | XX |
| $[0, 7]$ | $\{0\} \bigcup [1.5, 7]$ | $[5, 20]$ | | $[0, 4]$ | $[3, 8]$ | $\{0, 1\}$ | $[5, 20]$ |

digits one is defined by a sequence MPXX where M is the maximum ordinate of the camber line in hundredth of chords[1], P is the position of this maximum from the leading edge in tenth of chords and XX the maximum thickness in hundredth of chords. For the 5 digits series, each airfoil is defined by a sequence LPQXX. Digits L and P define in a more sophisticated manner than the 4 digits sequence the maximum camber of the camber line, Q is a boolean that switches between a single-cambered airfoil and a double-cambered one which allows in the latter case to achieve a theoretical pitching moment of 0. The last two digits XX have the same definition as in the 4 digits case.

Each simulation is first defined by an airfoil drawn in the 4 and 5 digits series families. The sampling strategy in those two series is given in Table 1. In our previous work [4], we chose to use the UIUC Airfoil Database [52] to build the dataset but we decide here to restrict our airfoil design space to the NACA 4 and 5 digits series. Those series are already rich families of airfoils that have been widely used historically and they are easier to handle for the automation of the mesh generation due to their explicit parametrization. Moreover, in the 4 digits series, we choose to sample the parameter P between 0 and 7 and to set the drawn parameters in the interval $(0, 1.5]$ to 0. We motivate this choice as airfoils with P in the range $(0, 1.5]$ have their maximum camber close to the trailing edge which can lead to unusable airfoils.

Examples of different airfoils, details on the generation of such airfoils, and empirical statistics of the drawn parameters are given in Appendix H.

**Mesh generation.** As airfoils are pretty simple geometries, we use the multi-block hexahedral mesh generator *blockMesh* from OpenFOAM v2112 [64] to mesh our shapes. We build a C-Grid mesh for each airfoil, mimicking the mesh developed by NASA for the NACA 0012 and 4412 cases [10]. Boundaries are at 200 chords of the airfoil to reduce the impact of boundary conditions on simulations. In Figure 1, we show the different block definitions with an example of a ready-to-use mesh and a final mesh on a classical airfoil. As we aim for accuracy in the computation of the global forces over the airfoil surface, such as the wall shear stresses, we mesh the boundary layer such that the first cells of the surface are of height $2\,\mu m$ leading to a $y^+$ of around 1 in the worst case of our design space. This leads to meshes from $250\,000$ to $300\,000$ cells. All the technical details and definitions of the meshing procedure are given in Appendix I.

**Dataset generation.** For the generation of the dataset, we run 1000 simulations, each defined by an airfoil, a Reynolds number, and an angle of attack. We choose to only run 1000 simulations as one of the goals of this dataset is to be close to real-world settings, *i.e.* limited quantity of data. The airfoil is sampled from the distribution given in Table 1. We motivate the design space of the initial conditions to reproduce the panel of flight conditions encountered in subsonic flights. We stop at a Mach number of 0.3 (Reynolds number of roughly 6 million) to keep the incompressible assumption valid and we start at a Reynolds number of 2 million as it is a correct lower bound of flight velocity (around 60 knots). The lower bound for angles of attack, $-5°$, is chosen such as cambered airfoils have a lift coefficient of roughly 0 and the upper bound, $15°$, is chosen to prevent stall and unsteady patterns in the trail of airfoils. Those ranges are tighter than the one chosen in our previous work [4] but better represent the classical ranges of velocity and angle of attack encountered in subsonic flight conditions. We then run the simulations with the help of the steady-state RANS solver *simpleFOAM* via the SIMPLEC algorithm [8, 16] and with the $k - \omega$ SST turbulence model [42] until convergence of drag and lift coefficients. Simulations are done on 16 CPU cores of an AMD Ryzen™ Threadripper™

---

[1]One chord is the characteristic length of the airfoil, in our case $1\,m$

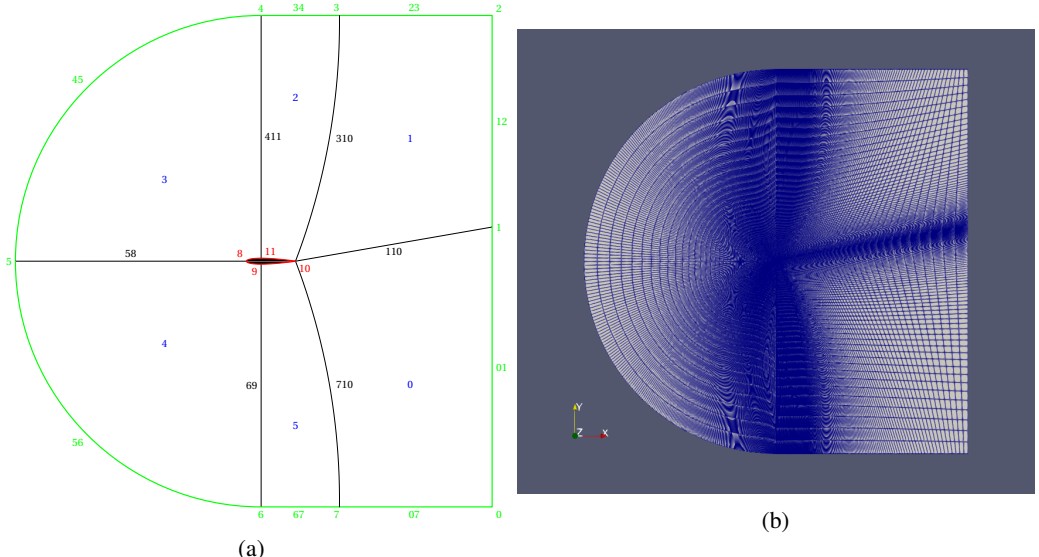

(a)

(b)

Figure 1: Example of mesh for the NACA 0012 at an angle of attack of $10°$. (a) Scheme of the multi-block mesh. Point number 1 moves following the angle of attack; all other points are fixed. The contour in red (airfoil) and green (freestream) are the domain's boundaries. (b) The entire domain of a ready-to-use mesh.

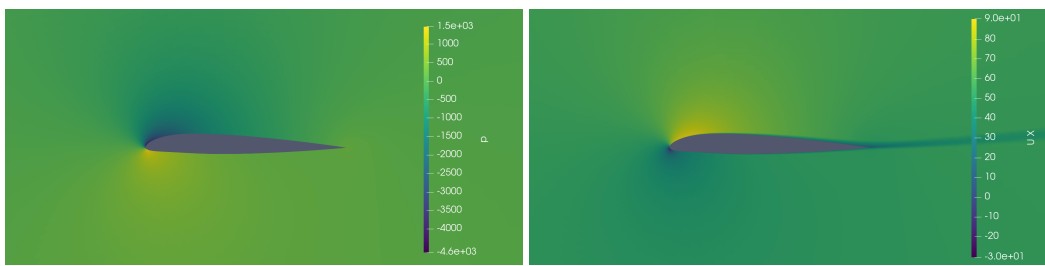

Figure 2: Example of a pressure (left) and $x$-component of the velocity (right) fields around a NACA $(2.123, 3.832, 1, 9.902)$ at a velocity of $54.238\,\mathrm{m\,s^{-1}}$ and at an angle of attack of $7.911°$.

3960X. Figure 2 shows a near view of the pressure and $x$-component of the velocity fields. Boundary conditions types and values for each simulation are given in Appendix J.

**Force coefficients.** One of the important quantities when simulating the fluid flow around a geometry is the force acting on it (see chapter 4 of [2]). This force is made by the contribution of the pressure and the viscous stresses at the surface of the geometry (called wall shear stress). The force component collinear to the free-stream velocity is called the drag $D$, and the one orthogonal to the free-stream velocity is the lift $L$. If divided by $q_\infty := \rho U_\infty^2 A/2$, where $\rho$ is the fluid specific mass, $U_\infty$ the inlet velocity and $A$ the characteristic area of the geometry (in our case we take the chord as the 1D surface characteristic "area", *i.e.* $A = 1\,\mathrm{m^2}$), those components give the dimensionless drag coefficient $C_D := D/q_\infty$ and lift coefficient $C_L := L/q_\infty$. Other quantities such as the pitching moment can also be computed with force acting on the body, but we will only focus on the drag and lift coefficients in this work. To compute the wall shear stress, we need to compute the velocity gradient at the surface of the airfoil. We compute it discretely with the help of the *gradient filter* in ParaView [3] over the mesh.

## 4  Benchmarking Setup

The machine learning task consists in predicting the different spatial fields such as the mean-field velocity and reduced pressure $\bar{u}$ and $\bar{p}$ and the force acting over airfoils. The turbulent viscosity $\nu_t$ is

not mandatory in the regression task as we do not need it to compute the force over an airfoil[2]. Still, it can give insights into the local intensity of turbulence in the volume. Regarding the force coefficients, and more precisely, the drag and lift coefficients, from a design process standpoint, we are more interested in the rank correlation with the true values than with the Mean Squared Error (MSE). Indeed, if the rank of the coefficients is accurately approximated, the optimization process will lead to the same airfoil. We can also add linear correlation plots for qualitative prediction information. Moreover, we are dealing in this dataset with hexahedral meshes with more than 250.000 cells which represent roughly the same number of nodes. These are already big meshes for 2D simulations but it is nothing compared to 3D cases where the number of cells can be of tens of million. To train a model on such simulations, we need to find a way to reduce the numerical complexity of the problem. Cropping close to the geometry is one way to do so, but as most cells lie in the vicinity of the geometry, it is not sufficient. Also, as we want to infer the force acting over the airfoil accurately, we cannot treat the cropped simulation as an image and work on a sub-sampled regular grid.

In this work, we choose the strategy which consists of regressing the four fields $\bar{u}_x$, $\bar{u}_y$, $\bar{p}$ and $\nu_t$ and compute the wall shear stress and the associated forces as post-processing to follow the form of RANS equations. We present a method to take into account the remarks mentioned above. Finally, we use the 1000 simulations to build four different setups:

- *Full data regime:* 800 samples for the training set and 200 for the test set

- *Scarce data regime:* 200 samples for the training set and the same test set as in the full data regime

- *Reynolds extrapolation:* the training set is composed of the samples with a Reynolds of three to five million, and the test set is formed by the samples with a Reynolds of two to three and five to six million

- *Angle of attack extrapolation:* the training set is composed by the samples with an angle of attack between $-2.5°$ and $12.5°$ and the test set is composed by the samples with an angle of attack from $-5°$ to $-2.5°$ and $12.5°$ to $15°$.

**Preprocessing.**    As we do not need the far-field to get rid of the boundary conditions impact on the simulation as in CFD, we crop all the simulations to a rectangle of size $[-2, 4] \times [-1.5, 1.5]$ meters. It allows us to limit our point clouds' size and make the network focus on the interesting part of the simulations. Moreover, data normalization is important in DL to make the optimization process easier or feasible. We use normalization with the means and the standard deviations of the training set field components.

**Loss, sampling, and graph construction.**    At the end of the cropping procedure, we still have to deal with roughly 150.000 cells in each simulation giving about the same number of nodes. To train a model on such simulations, we need to find a way to reduce the numerical complexity even more.

To handle this numerical complexity, at each epoch, we choose to sample uniformly on the cropped mesh 32.000 nodes and, when necessary, reconstruct a radius graph of radii $5\,\mathrm{cm}$ with a maximum number of neighbors of 64[3]. This approach has several advantages, it allows us to directly control the numerical complexity with the number of sub-sampled nodes, the radii of the graph, and the maximum number of neighbors inside it. Moreover, during the inference, it is straightforward to infer fields on each node of the initial mesh by making multiple forward passes with different sub-sampling until every node has been seen and averaging the outputs on the nodes that have been seen multiple times. It allows us to compute the velocity gradient on the airfoil with the help of the initial mesh and ParaView's pythonic interface PyVista [57] to be able to compare with the surface force targets. However, one downside effect of the method is that the mesh densities bias the learning procedure, and the learned models may not generalize well to different types of meshes. In this dataset, we are not facing this problem as all the meshes are generated via the same procedure. Additionally, we can not sample independently on the airfoil and on the volume to bias models to be more accurate on the airfoil as the force highly depends on the pressure field at the surface. Finally, The loss $\mathcal{L}$ used in this

---

[2]the boundary condition of the turbulent viscosity on the airfoil is 0 in our case

[3]which means that we connect nodes only very locally compared to the characteristic length of $1\,\mathrm{m}$ of airfoils

work is the sum of two terms, a loss over the volume $\mathcal{L}_\mathcal{V}$ and a loss over the surface $\mathcal{L}_\mathcal{S}$:

$$\mathcal{L} := \underbrace{\frac{1}{|\mathcal{V}|}\sum_{i\in\mathcal{V}}\|f_\theta(x_i) - y_i\|_2^2}_{\mathcal{L}_\mathcal{V}} + \underbrace{\lambda\frac{1}{|\mathcal{S}|}\sum_{i\in\mathcal{S}}\|f_\theta(x_i) - y_i\|_2^2}_{\mathcal{L}_\mathcal{S}} \tag{3}$$

where $\mathcal{V}$, $\mathcal{S}$ are respectively the set of the indices of the nodes that lie in the volume and on the airfoil, $x_i \in \mathbb{R}^7$ is the input at node $i$ containing the spatial coordinates, the inlet velocity, the Euclidean distance function between the node and the airfoil, and the unit surface outward-pointing normal for points on the airfoil (filled with zeroes otherwise). The targets $y_i \in \mathbb{R}^4$ at node $i$ contain the velocity, the pressure, and the turbulent kinematic viscosity at this node. And $f_\theta$ represents the model used. The coefficient $\lambda$ is used to balance the weight of the error at the surface of the geometry and over the volume[4]. We have to emphasize that this loss is not necessarily a good proxy when it comes, for instance, to infer the wall shear stress accurately or ensuring that the inferred fields satisfy the RANS equations.

**Metrics and visualizations.**   One of the challenges of this dataset is to build models that manage to predict the form of the boundary layer accurately. To evaluate the performance of the models, we define qualitative and quantitative metrics.

To qualitatively check this accuracy, we propose to plot the components of the velocity and the turbulent viscosity (if regressed) in the boundary layer of airfoils at different chord lengths. Also, we propose to check the accuracy of the prediction for the pressure and skin friction coefficients on the airfoil as they carry important information on the wing's behavior in flight conditions. Finally, we propose to plot the predicted force coefficients with respect to the true coefficients which gives us qualitative information about the correlation between both variables.

In terms of quantitative metrics, we use the MSE for each field on the volume and over the airfoil to measure the accuracy of our models. Moreover, we compute the mean and standard deviation of the relative error on the drag and lift coefficients. Finally, we compute Spearman's rank correlation coefficient between the true and predicted force coefficients. From a design process point of view, the last coefficient is the most crucial quantity to maximize as it quantifies the monotonic correlation between the true and predicted force coefficients. If this coefficient is close to 1, we can expect our model to be able to find the best airfoil maximizing or minimizing the chosen force coefficient even if the inferred value is not close to the true value. The experiments have been done on an NVIDIA RTX 3090 24Go.

## 5   Benchmarking Results

To propose baselines for the problem, we train three standard GDL models and a Multi-Layer Perceptron (MLP) in the full data regime. The associated results for the three other tasks are given in Appendix M. Each model is preceded by an encoder and followed by a decoder, both defined by a MLP and trained together with the model. We follow the setup defined in the previous section for the training and testing procedures. Each model is trained 5 times to compute a mean and a standard deviation for the different metrics. For each metric, we bold the best-performing method. We choose as baselines a GraphSAGE [27], a PointNet [11], a Graph U-Net [23] and a MLP. Those baselines have been chosen as they access different types of information. The MLP only has access to the features of the nodes, the GraphSAGE has in addition access to local neighborhood information, the PointNet conditioned a deep MLP with global features, and the Graph U-Net access from local to global neighborhood information via its multi-scale architecture. Models are trained in the same conditions and the details of architectures and hyperparameters can be found in Appendix L.

In Table 2, we give the MSE over the volume and at the surface of airfoils for the different regressed fields. In Table 3 we give the mean relative errors on the force coefficient and the Spearman's rank correlation coefficient. In Table 4 we compare the computational time to run a simulation, train a model and infer on a new example. In Figure 3 we plot the predicted force coefficients with respect to the true coefficients. Plots of the velocity and turbulent viscosity profiles in the boundary layer and surface coefficients for randomly chosen test geometries are given in Appendix M.

---

[4]In this work $\lambda$ is set to 1.

Table 2: Mean squared error on the different normalized fields for a MLP and standard GDL baselines on the full data regime test set. Only the reduced pressure is given on the surface as the other quantities are null via the boundary conditions. Those quantities are directly regressed by the models.

| Model | Volume | | | | Surface |
|---|---|---|---|---|---|
| | $\bar{u}_x$ ($\times 10^{-2}$) | $\bar{u}_y$ ($\times 10^{-2}$) | $\bar{p}$ ($\times 10^{-2}$) | $\nu_t$ ($\times 10^{-2}$) | $\bar{p}$ ($\times 10^{-1}$) |
| MLP | $0.95 \pm 0.06$ | $\mathbf{0.98 \pm 0.17}$ | $0.74 \pm 0.13$ | $1.90 \pm 0.10$ | $1.13 \pm 0.14$ |
| GraphSAGE | $\mathbf{0.83 \pm 0.01}$ | $0.99 \pm 0.05$ | $\mathbf{0.66 \pm 0.05}$ | $1.60 \pm 0.21$ | $0.66 \pm 0.10$ |
| PointNet | $3.50 \pm 1.04$ | $3.64 \pm 1.26$ | $1.15 \pm 0.23$ | $2.92 \pm 0.48$ | $0.93 \pm 0.26$ |
| Graph U-Net | $1.52 \pm 0.34$ | $2.03 \pm 0.39$ | $0.66 \pm 0.08$ | $\mathbf{1.46 \pm 0.14}$ | $\mathbf{0.39 \pm 0.07}$ |

Table 3: Relative errors (Spearman's rank correlation) for the predicted drag coefficient $C_D$ ($\rho_D$) and lift coefficient $C_L$ ($\rho_L$). We want Spearman's correlation to be close to one. Those quantities are computed as a post-processing from the unnormalized regressed fields.

| Model | Relative error | | Spearman's correlation | |
|---|---|---|---|---|
| | $C_D$ | $C_L$ | $\rho_D$ | $\rho_L$ |
| MLP | $4.289 \pm 0.679$ | $0.767 \pm 0.108$ | $-0.117 \pm 0.256$ | $0.913 \pm 0.018$ |
| GraphSAGE | $\mathbf{4.050 \pm 0.704}$ | $0.517 \pm 0.162$ | $-0.303 \pm 0.124$ | $0.965 \pm 0.011$ |
| PointNet | $14.637 \pm 3.668$ | $0.742 \pm 0.186$ | $-0.022 \pm 0.097$ | $0.938 \pm 0.023$ |
| Graph U-Net | $10.385 \pm 1.895$ | $\mathbf{0.489 \pm 0.105}$ | $-0.138 \pm 0.258$ | $\mathbf{0.967 \pm 0.019}$ |

The Graph U-Net model significantly outperforms other models for the pressure at the surface, this correlates with its performance on the relative error and Spearman's correlation for the lift coefficient. However, it struggles to learn the velocity field compared to local models such as GraphSAGE and MLP. The GraphSAGE model seems to be a good trade-off, in this setting, between complexity and performance as it achieves almost equivalent performance with the Graph U-Net, is almost twenty times faster to call, and has half of the number of parameters of the Graph U-Net.

From the plots of the different examples of boundary layers and skin friction coefficients given in Appendix M, we conclude that the models have difficulties to predict the wall shear stresses as the velocity values at the closest nodes from the geometry are often largely overestimated. This particularly affects the accuracy of the drag coefficient as we can see with the Spearman's correlation $\rho_D$, the relative error on the drag coefficient, and the plot of the predicted drag with respect to the true drag coefficients (see Figure 3 left). However, the wall shear stress has a small impact on the lift coefficient compared to the pressure at the surface of airfoils. Hence, as the pressure is more accurately inferred compared to the wall shear stress, as we can see by looking at the plots of the pressure coefficient at the surface of airfoils in Appendix M, the inferred lift coefficient is also more

Table 4: Running time for one simulation on 16 CPU cores of an AMD Ryzen™ Threadripper™ 3960X compared to training and inference time of the different models on an NVIDIA GEFORCE RTX 3090. The inference time is given for one call of a model on a batch of 32000 nodes, for one simulation we need around a hundred calls to get a result on the entire mesh as the nodes are chosen randomly on the CFD mesh. The number of parameters for each model is given as additional information.

| Model | Running time | | # Parameters |
|---|---|---|---|
| | Training | Inference (μs) | |
| MLP | $\sim 2\,\mathrm{h}20\,\mathrm{min}$ | $13.3 \pm 0.2$ | 19988 |
| GraphSAGE | $\sim 4\,\mathrm{h}20\,\mathrm{min}$ | $20.9 \pm 2.3$ | 29204 |
| PointNet | $\sim 2\,\mathrm{h}40\,\mathrm{min}$ | $33.9 \pm 3.5$ | 75244 |
| Graph U-Net | $\sim 6\,\mathrm{h}50\,\mathrm{min}$ | $357.8 \pm 36.9$ | 65820 |
| Simulation | | $\sim 25\,\mathrm{min}$ | |
| Dataset | | $\sim 20\,\mathrm{days}$ | |

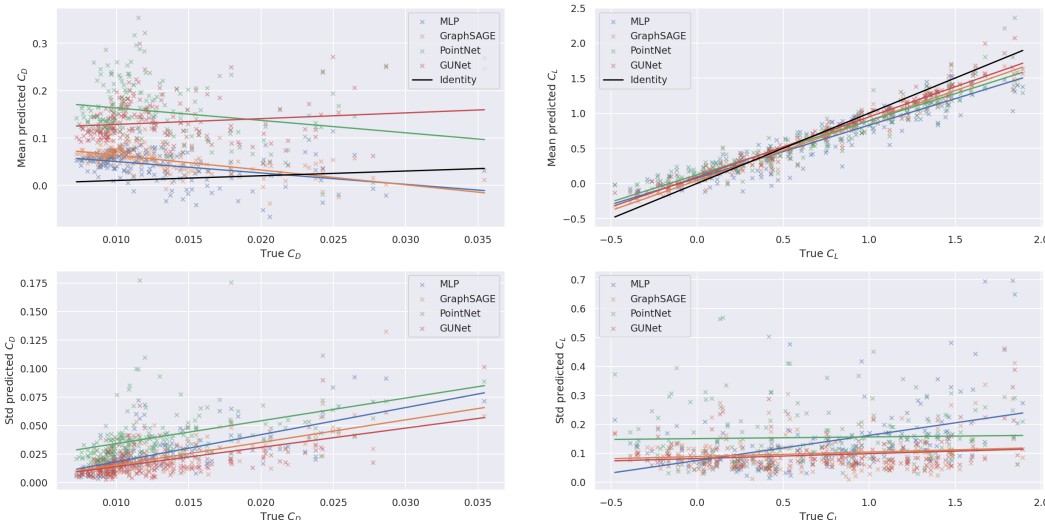

Figure 3: Predicted drag (left) and lift (right) coefficients with respect to the true ones. The mean (top) and standard deviation (bottom) of each point on the five copies of the trained models are separated for sake of readability. Linear regression is done for each point cloud to highlight linear trends. On the top plots, the Identity graph is given in black for comparison.

Table 5: Mean squared error on the different normalized fields, relative error, and Spearman's correlation for the force coefficients on the four different tasks for the GraphSAGE model. Only the reduced pressure is given on the surface ($\bar{p}_s$) as the other quantities are null via the boundary conditions.

| Field / Coefficient | Task | | | |
|---|---|---|---|---|
| | Full | Scarce | Reynolds | AoA |
| $\bar{u}_x$ ($\times 10^{-2}$) | $0.832 \pm 0.015$ | $1.457 \pm 0.125$ | $7.558 \pm 1.046$ | $4.435 \pm 0.334$ |
| $\bar{u}_y$ ($\times 10^{-2}$) | $0.994 \pm 0.052$ | $1.454 \pm 0.123$ | $3.498 \pm 0.613$ | $9.400 \pm 2.167$ |
| $\bar{p}$ ($\times 10^{-2}$) | $0.661 \pm 0.050$ | $4.696 \pm 0.804$ | $3.826 \pm 0.248$ | $10.908 \pm 2.164$ |
| $\nu_t$ ($\times 10^{-1}$) | $0.160 \pm 0.021$ | $0.611 \pm 0.079$ | $1.694 \pm 0.383$ | $5.178 \pm 0.365$ |
| $\bar{p}_s$ ($\times 10^{-1}$) | $0.662 \pm 0.103$ | $1.945 \pm 0.336$ | $1.797 \pm 0.338$ | $7.638 \pm 0.945$ |
| $C_D$ | $4.050 \pm 0.704$ | $3.504 \pm 0.998$ | $8.971 \pm 1.278$ | $5.589 \pm 1.090$ |
| $C_L$ | $0.517 \pm 0.162$ | $0.385 \pm 0.097$ | $0.616 \pm 0.124$ | $0.818 \pm 0.300$ |
| $\rho_D$ | $-0.303 \pm 0.124$ | $-0.139 \pm 0.175$ | $0.013 \pm 0.064$ | $0.055 \pm 0.171$ |
| $\rho_L$ | $0.965 \pm 0.011$ | $0.981 \pm 0.006$ | $0.927 \pm 0.027$ | $0.908 \pm 0.019$ |

accurately inferred and the rank is better predicted (see Figure 3 right) leading to a Spearman's correlation close to one.

**Difficulties of the different tasks.** In Table 5 we give the scores of the GraphSAGE model on the four different tasks. The scores are given on the test set associated with each task. This makes difficult the direct comparison as the test set for the Reynolds and angle of attack extrapolation regimes are both different from the test set of the full and scarce regimes.

In the full and scarce regime, the MSE over the different fields shows that the GraphSAGE model is performing better in interpolation when more data is available, as expected. However, the scores on the force coefficients are slightly higher in the scarce regime which tells us that the MSE over the different fields is not necessarily a good proxy for the accuracy of the force coefficients. This can be understood as the computations of the force coefficients involve an integration over the surface and can lead to the accumulation or compensation of local errors.

In the Reynolds and angle of attack extrapolation regimes, the MSE scores are significantly higher than in the full and scarce data regimes. As expected, the extrapolation tasks are more difficult than

the interpolation tasks. The Spearman's correlation for the lift coefficient is also significantly lower for both extrapolation tasks than for the interpolation ones, supporting the previous observation.

Finally, in Table 4, we confirm that even for a two-dimensional case, the training cost of models is rapidly amortized (after, in the worst case, a dozen of simulations).

# 6    Conclusion

In this work, we presented a high-fidelity dataset of solutions of the two-dimensional RANS equations around NACA airfoils. Simulations have been done at Reynolds of the order of magnitude of what we find in subsonic flight regimes mimicking classical aerodynamics setups. We defined four ML tasks highlighting the different challenges of surrogate models, from scarce data regimes to extrapolation. We proposed different metrics focusing not only on the velocity and pressure fields but also on the force coefficients. Those metrics quantify the ability of ML models to accurately predict fields and force coefficients in addition to their ability to rank the latter, for example, for shape optimization. Different baselines have been introduced from the GDL framework, highlighting the need for models that can handle unstructured point clouds in order to be able to accurately predict force coefficients. Those baselines have shown in the proposed setting, as expected, that the prediction of the drag coefficient is more challenging than the prediction of the lift coefficient as the wall shear stress is derived from the velocity field and not directly regressed like the pressure.

**Metrics hierarchy.**    From a design-oriented perspective we may set a hierarchy for the proposed metrics. The Spearman's correlation for the force coefficients is the main metrics to maximize the recovery of the best airfoils in terms of lift-over-drag ratio as it quantifies the ability of models to preserve the force coefficient ranks. In addition to this metric, the plots of the predicted with respect to true force coefficients give qualitative information on the accuracy of the model for each simulation of the test set. Then, the relative errors for the force coefficients are important but not crucial from a design perspective and their minimization is secondary. We may say that a model is effective if it has Spearman's correlations close to one and accurate if it has low relative errors and low MSE on the fields over the volume and the airfoil. The goal here is an effective and accurate model. For relative errors, the bound of $5\%$ is often used to state that a model is accurate enough.

**Limitations.**    Concerning the dataset itself, we restricted the design space of airfoils to NACA 4 and 5 digits for the sake of simplicity and meshing automation but we can expect that models trained on this dataset will have difficulties generalizing to more exotic shapes. Following this, we did not propose an extrapolation test on out-of-distribution airfoils. Also, even though the problem proposed is a classical aerodynamics one, it is two-dimensional and does not reflect entirely the complexity of three-dimensional natural phenomena which implies that models working on this dataset could not necessarily be extended to more generic three-dimensional cases.

In terms of baselines, we proposed four architectures of different types, an MLP, a GNN, a network acting on point clouds, and a multi-scale GNN architecture. The GNN approach suffers from its heaviness when dealing with large graphs leading to the necessity of building a downsampling strategy. In addition to that, auto differentiation with respect to positions is not possible with GNN as an entire graph is given in inputs. This leads to difficulties in the inference process and the need to rely on input CFD meshes to compute derivatives using numerical schemes. On the other hand, the MLP approach allows more flexibility and does not require a subsampling strategy or the input CFD mesh to compute derivatives. Uniform sampling is also possible in this case and several models have already shown their ability to fit complex signals [54, 39]. The main downside of such approaches is the generalization capacity requiring conditioning or hypernetworks to work on multiple examples and generalize to unseen ones. Moreover, let us mention equivariant networks [5] as another promising direction to handle the lack of data often encountered in such tasks by leveraging symmetries of the problem. Finally, neural operators handling unstructured data such as DeepONet [41] are by definition well suited for such tasks and could be mixed with previous techniques to achieve high-performance surrogate models.

In total, we consider this work as a first step towards the generic treatment of real-world physical phenomena in ML. We hope that it will lower the potential barrier for entering the field of ML applied to physical systems and that it will encourage the construction of models that are not only good on the predicted fields but also on meaningful derived quantities.

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
