# OpenReview forum: "AirfRANS: High Fidelity Computational Fluid Dynamics Dataset for Approximating Reynolds-Averaged Navier–Stokes Solutions"
_NeurIPS.cc/2022/Track/Datasets_and_Benchmarks — NeurIPS 2022 Datasets and Benchmarks _

### Official Review · Reviewer_t2W6 · 2022-07-06
**Good content but lack of clarity**

**Rating:** 6
**Confidence:** 3

**Strengths:**

Procedures for the simulations are well-documented, sufficient references and literature reviews of state-of-the-art, and detailed descriptions of the simulation setups and theoretical background are provided as supplementary materials.

**Weaknesses:**

Models chosen as baselines only include purely ML models. There are many new hybrid ML models that were recently proposed to tackle turbulent flow problems (see e.g. TF-Net, Wang et. al., 2020), which I think could be more useful and give more insights.
The writing in general could be improved significantly. This also causes lack of clarity in most of the content.
Link to Github repository doesn't work.
The authors also provide baseline for full and scarce data regime, also for extrapolation, but the comparison is not made very clear.

**Additional Feedback:**

I would be happy to change my review if the quality of writing is significantly improved.

**Clarity:**

Quality of writing is poor. There are many sentences that are not understandable and confusing, because of grammatical errors.

**Correctness:**

All the statements claimed in the paper seem to be consistent and sound.
My question is: is all the simulation setup used to generate the dataset representative enough to cover various scenarios of fluid flow around airfoils? More concretely, are the boundary conditions and parameter values used are wide enough?
Specifically about boundary condition: would a new model needs to be trained if a boundary condition is changed?

**Documentation:**

Procedures are well documented and all softwares that are mentioned to be used are open access.
However, the link to the Github repository can't be accessed.

**Relation To Prior Work:**

The related work section provides a sufficient literature review of state-of-the-art methods and models.
However, I'm wondering why the authors use only pure ML methods as baselines and not include physics-informed ML such as PINN, TF-Net (specifically developed for modelling turbulent flow) or operator based model such as FNO, DeepONet.
These models might be more useful as baselines compared to MLP or U-Net.

**Summary And Contributions:**

The authors provide high resolution dataset of the 2D Reynolds-Averaged-Navier-Stokes solution over airfoils, and compare 4 different deep learning models as baseline surrogates. The authors also test the generalization ability by extrapolating the simulation parameters for the test dataset. While this is an important and appealing topic in the computational fluid dynamics research field, the paper lacks clarity mainly on the motivation and objectives, which I think is caused by poor writing.

*EDIT: I improved my score from 4 to 6 after the rebuttal.

---

> ### Author Response · Authors · 2022-08-15
> **Response to Reviewer t2W6**
>
> We thank the reviewer for the comments and below we provide an answer to the underlined concerns and more clarifications about the paper and the goal of this work.
>
> Summary and Contributions:
>
> We are happy to read your support about this topic for ML & CFD research fields and we are very sorry that you found that the paper lacked clarity. We added a paragraph called “Design-oriented dataset.” in section 3 Dataset Presentation to clarify the motivations and objectives of the dataset. We also tried to correct typos and heavy sentences. We would be pleased to read your suggestions for increasing our quality of writing in order to improve the global quality of the paper.
>
> Weaknesses:
> 1. The GitHub repository to reproduce the experiments is now available here and we also created a GitHub repository to easily run new simulations or generate similar datasets as this one.
>
> 2. The scores of the GraphSAGE model have been added in Table 5 in section 5 Benchmarking Results for all of the four tasks and a comment of those scores is given in the same section onat lines 277 to 290. We also added the scores of all the models on the four tasks in Table 16 and 17 in section M Additional Results of the supplementary materials in addition to the associated plots and visualizations.
>
> 3. The TF-Net is a network working on voxels or pixels. In the proposed setting, one of the main problems is to accurately predict the force coefficients acting over airfoils. Hence, we must work on unstructured data and networks such as TF-Net can not handle it.
>
> 4. We insist that it is not the purpose of this work to provide an exhaustive benchmark but instead to make available a well designed dataset with appropriate evaluation protocols to explore the capabilities of DL to solve real-world problems in CFD. Our goal is then explicitly to encourage researchers working on this topic to evaluate their own methods onto this challenging dataset. There are currently many approaches combining physics and machine learning for solving this family of problems. We argue that current evaluation protocols are most of the time far too simple, while the one proposed here is representative of several of the difficulties and challenges of CFD.
>
>  Correctness:
> 1. The simulation setup and the design space chosen is representative of flight conditions in a subsonic low Reynolds regime. The lower bound of the Reynolds number (2 millions) corresponds to a velocity of roughly 60 knots which is a correct lower bound in flight conditions. The upper bound (6 millions) is chosen such that the incompressibility assumption stays correct and the simulations close to real dynamics. Concerning the angle of attacks, the lower bound of -5 degrees is chosen such that the cambered airfoils have a lift coefficient of roughly 0 and the upper bound of 15 degrees is chosen to prevent stall and unsteady behaviors in the trail of airfoils. This is also a classical range of angle of attacks in flight conditions.
>
> 2. We are not sure to correctly understand the question. Surrogate models are built to interpolate or extrapolate around initial conditions or geometry design spaces, in that sense, a model trained on this dataset will not need to be retrained on another initial condition to give a good approximation of the associated steady-state simulation. At least, it is the goal of the proposed tasks to build such a model. It is, for example, different from PINNs that are trained on one and only one dynamics and need to be retrained for each new boundary or initial condition.
>
> Relation to prior work:
>
> We tried to propose simple and historical networks coming from different backgrounds. The MLP is the classical baseline for every deep learning task, the GraphSAGE is a standard architecture and single-scale GDL baseline, the PointNet adds a global information to a deep MLP and acts on unstructured data and the Graph U-Net is also a standard multi-scale GDL baseline. As our goal is to present the dataset and not to run an exhaustive benchmark on it, we found it sufficient to stick with classical baselines. Moreover, the size of each simulation makes the application of physics-based architectures more challenging and sometimes unfeasible. For example, the neural operators GNO or MGNO (cite) could have been a good baseline for this task as we already used it in our previous dataset paper but it was too greedy in memory to run on this one. Moreover, we need neural networks that achieve learning on unstructured data, the TF-Net and the FNO act on regular grids which is unsuitable for our case. Also, PINNs are not designed for surrogate modeling and are not suited for our case. Finally, the DeepONet network is a neural operator that could have been benchmarked in this paper, we added a comment on it at the end of the section 6 Conclusion (lines 323 to 327). We keep it for a subsequent work.

---

> > ### Comment · Reviewer_t2W6 · 2022-08-17
> > **Clarification improved the paper**
> >
> > Dear Authors, thank you for addressing my comments and concerns.
> >
> > Here are some of my further comments:
> > - Github repository is now available and the code seems to be structured very well and the API is quite simple.
> > - Comparison between different tasks are now presented in the paper and provides more clarity to the content.
> > - Objectives of the paper are now explicitly written in the introduction and the dataset presentation section, which also improve the clarity.
> >
> > Based on these points alone, I increased my score from 4 to 5. I think your rebuttal is excellent, and I am willing to further increase my score, if you some important clarifications also in the paper (not only here), which I think would help the readers to better understand your work. Here are some of the issues that I find important to be put in the paper:
> > - Your answer to the relation to prior work. I particularly like your comprehensive answer here, and I think it would fit very nicely in Section 2. I understand that performing benchmark on everything will be impossible, but providing arguments on why these SotA models do not fit your purpose would be beneficial also for the readers to save their time from performing pointless experiments. Also, as far as I understand, PINN is discretization-independent but does not fit your purpose, but as far as I understand, FNO could also be independent from the discretization, since the inputs are only the previous state (at a certain point in space), and the grid coordinate (and the model itself do not receive any information about spatial correlation between neighboring cells). But please correct me if I am wrong.
> > - You wrote about the purpose of the work (airfoil design optimization), but how does your benchmark results relate to achieving this goal? A short explanation or summary in the conclusion would be nice.
> > - I still find the result section a bit lacking in explanations of the plots and tables (e.g. Table 2-4 and Figure 3). I also think it would be nice to put more emphasize on the metrics which are physically meaningful and provide a brief analysis on them.
> > - The range of the simulation parameters are also nicely explained in the rebuttal, but would also be better to be put in the paper.

---

> > > ### Author Response · Authors · 2022-08-24
> > > **Response to Reviewer t2W6**
> > >
> > > We thank the reviewer for their encouragement and for the proposition to increase the overall clarity of the paper. In the following, we give the details of the update and the answers to the underlined concern.
> > >
> > > 1. We added a short comment on the FNO (lines 74 to 75) and the PINN (lines 82 to 83) to emphasize the incompatibility between those networks and the proposed tasks. You are right concerning the PINN as it is defined in the original paper from Raissi et al. as those architectures are built to learn one and only one PDE. However, we could design a hyper-network or conditioning that allows PINN to generalize to multiple PDE. We explicitly gave the PDE solved in the simulations in the main paper as it allows us to easily add a loss to minimize the PDE residual in the training process when the architecture allows it. For example, we could have used it in the training of the MLP. Concerning the FNO, we would say that it is the use of the Fourier transform and more specifically the Fast Fourier Transform algorithm which makes this architecture unusable in our case. As we do not have a rectangular domain, it is not straightforward to effectively compute the Fourier transform of our signal.
> > >
> > > 2. Thank you for raising this question. We added a paragraph “Metrics hierarchy” (lines  327 to 336) in the conclusion to treat this question.
> > >
> > > 3. We added a short comment (lines 281 to 286) to comment on Table 2-4. We think that a discussion about Figure 3 is already given in section 5 (lines 287 to 297).
> > >
> > > 4. We added two comments in section 3 (lines 134 to 141 and lines 157 to 164) for this purpose.

---

> > > > ### Comment · Reviewer_t2W6 · 2022-08-26
> > > > **Response to Authors**
> > > >
> > > > Dear Authors,
> > > >
> > > > Thank you for addressing my concerns in a very thorough manner. I have again raised my score to 6.

---

### Official Review · Reviewer_EXMb · 2022-07-27
**Computational fluid dynamics airfoil dataset & benchmark for ML surrogate models**

**Rating:** 8
**Confidence:** 2
**Clarity:** The paper is overall well-written.

**Strengths:**

1. The dataset proposed seems sensible and well designed. From what I can see, care is put into construction and running of the simulations, as well as selection of the design space. The proposed dataset splits are also sensible. Overall, I think this is a very solid contribution and the dataset will be a useful benchmark for the field.

**Weaknesses:**

1. In the paper, four different dataset splits are described yet only the benchmark results for two of them (the interpolation regimes with "full data" and "scarce data") are provided. The extrapolation performance of ML models are key measurements and the results should be provided. Failures in extrapolation can also reveal unintentional flaws in the design space of the dataset so I would like to see those benchmarks to assess the construction of the dataset.


**Additional Feedback:**

1. For the 4-digit NACA airfoils, it is not clear to me what geometrical issues resulting from P < 1.5 are such that setting P to 0 would solve them. Is it an issue with mesh generation?

2. Could you explain what do you mean by "relative error" in the benchmarking results, and why these relative errors are lower in the scarce data regime than in the full data regime? eg 2.95 vs 3.92 for the MLP model.

**Correctness:**

The main claim of the paper is correct. The dataset is well-designed and the benchmarking done rigorously, however only results for two of the proposed four dataset splits are provided.



**Documentation:**

The dataset construction is clearly documented and available for download in the submission. The benchmarking is described in detail and a link to a GitHub repository is given, however the repo seems to be private/unavailable so I could not confirm reproducibility.

**Ethics:**

I don’t see major ethics concerns with this work.

**Relation To Prior Work:**

The authors clearly discuss how the dataset in this work is at a higher fidelity and covers a wider design space than previous contributions.

**Summary And Contributions:**

This paper proposes a dataset of fluid dynamics simulations of two-dimensional airfoils to serve as a benchmark of ML surrogate models for fluid dynamics. The dataset is constructed to cover a range of airfoil shapes and fluid flow conditions, and multiple physically relevant dataset splits are proposed. The benchmark results of several popular ML models are included.

The main contribution of this paper is the fidelity of the dataset itself, as well as design process for constructing the dataset.

---

> ### Author Response · Authors · 2022-08-15
> **Response to Reviewer EXMb**
>
> We thank the reviewer for the very encouraging appreciation and for the support. We answer the comments below.
>
> Weaknesses:
> 1. Thank you for pointing it out, we added Table 5 in section 5 Benchmarking results to give the scores of the GraphSAGE model over the four tasks and we commented on it in the same section (line 277 to 290). Moreover, we added the scores of all the models over the four tasks and the associated plots and visualizations in the section M Additional Results of the supplementary materials.
>
> Relation to prior work:
> We would like to specify that the design space in the AirfRANS dataset is not bigger than the design space of our previous dataset paper submitted at the GTRL workshop of ICLR 2022
> . Actually, we restricted the class of airfoils to the NACA 4 and 5 digits series whereas we were using the UIUC bank of airfoils for the previous dataset. The NACA 4 and 5 digits series are already rich families of airfoils that have been widely used historically and they are easier to handle for the automation of the mesh generation due to their explicit parametrization. Moreover, we restricted the angle of attacks from -5 degrees to 15 degrees as it is the classical range of angle of attack in flight conditions whereas the previous dataset contained lower angle of attacks leading to unpractical scenarios.
>
> Additional Feedbacks:
> 1. We did not encounter any problem in the mesh generation procedure, however, we put this constrained as the NACA 4 digits airfoils with P < 1.5 have a maximum of camber really close to the leading edge which can generate unphysical airfoils.
>
> 2. The relative error is defined by abs((true - predicted)/true) as usual. The relative errors and Spearman’s correlation of the force coefficients for the different models are consistently higher in the full data regime than in the scarce data regime. This can be explained by the fact that the MSE over the different fields is not necessarily a good proxy of how the models will perform on metrics involving integral coefficients such as the drag and lift coefficients. Indeed, as we integrate the pressure and the wall shear stress over the surface of airfoils, errors can be accumulated or compensated through this sum. This kind of information can be qualitatively checked with the surface profile of the pressure and the wall shear stress as given for example in Figures 12 and 13 in section M Additional Results of the supplementary materials.

---

> > ### Comment · Reviewer_EXMb · 2022-08-21
> > **Improved manuscript**
> >
> > Thank you for addressing my comments and concerns.
> >
> > - I am pleased that GitHub repositories for both dataset construction and model benchmarking are now available. From visual inspection the code looks legible and complete, though I have not had the time to explicitly verify the reproducibility of the results.
> > - The missing sections on benchmarking models on all four dataset splits have now been added.
> > - Additional sections added to the paper have increased the overall clarity.
> >
> > I appreciate the provided clarification in your response regarding the prior work and explaining the metrics. I think including it in the paper (as well as additional clarifications on the simulation parameters/explanation of the result plots raised by other reviewers) would be a great addition for helping readers understand this work.
> >
> > A final comment:
> > - at what point would you define your benchmark to be "solved"? Would there be target values for the benchmark metrics (eg relative error < 0.1) which would indicate that a particular model should be "good enough" to replace running a simulation? I think this is an important point for researchers both within and outside of the field to gain an intuition on whether meaningful progress is being made.

---

> > > ### Author Response · Authors · 2022-08-24
> > > **Response to Reviewer EXMb**
> > >
> > > Again, we thank the reviewer for their comments. The question raised is difficult to answer and we hope to give a satisfying answer to it.
> > >
> > > In the perspective of pure shape optimization, relative errors are of marginal importance if the rank is preserved. We could think of a scenario where the model always adds an arbitrary high constant offset to the force coefficient prediction leading to arbitrary high relative errors and a Spearman’s correlation of one. The optimization process with this model will lead to the true optimum airfoil whereas it performs poorly in absolute values of the coefficients. However, we may still say that there is a hierarchy in the proposed metrics and that we actually would like a model that produces pertinent optimized airfoils and accurate field and coefficient predictions. In practical settings, we often set a bound of 5% on the relative errors to qualify a model as “accurate”. We added a paragraph “Metrics hierarchy” in the conclusion (lines 327 to 336) to discuss it. Thank you for your important question.
> > >
> > > We also added comments in the paper to discuss the choice of the design space (lines 134 to 141 and 157 to 164) and the scores of the different models on the full data regime task (lines 281 to 286).

---

### Official Review · Reviewer_cACM · 2022-07-27
**A dataset for studying the 2D RANS equations over airfoils but several sections in the paper are unfinished.**

**Rating:** 7
**Confidence:** 3
**Correctness:** The claims are correct. The evaluatio…
**Clarity:** Yes. It is well written and easy to f…

**Strengths:**

1) The authors have designed four different challenges with this dataset, including full data regime, scarce data regime, Reynolds extrapolation, and angle of attack extrapolation. The baselines with MLP, GraphSAGE, PointNet and Graph U-Net are also provided, which make it easy for other researchers to build new ML models and compare them with the results in this paper.
2) The experiment consists of both quantitative and qualitative evaluation metrics. The MSE and Spearman's rank correlation are calculated for velocity, pressure field and force coefficients to quantify the ability of ML models.
3) The paper has a detailed description of the dataset construction including the RANS equations, airfoil design, mesh generation, and dataset generation. It is easy for readers to understand the whole procedure. The computational challenges in the data generation due to the number of meshes are clearly clarified.

**Weaknesses:**

1) The discussion about the characteristics of this dataset is needed. Most of the paper is about the generation of the data, but the information about the feature of the dataset itself is missing. I think the authors need to explain why this dataset can be the typical benchmark for the design of ML models in this area.
2) Several sections about the networks and the dataset are not finished in the appendix. Only one baseline about the full data regime is provided in the paper while the other three are still unavailable. It is hard for me to give a fair evaluation of the benchmarking without these contents.


**Additional Feedback:**

1) Page 5 line 174 "the four field $\bar{u}$, $\bar{p}$ and ${v}_{t}$" Are the velocity fields $\bar{u}$ with different spatial component treated as two fields here.
2) What is the difference between the dataset with the dataset you present in the ICLR workshop？ Why call this dataset a high-fidelity version?
3) In Figures 1a and 2, the numbers in the images are too small to see.
4) The dataset is generated with random airfoil, Reynolds number, and angle of attack. The range of the Reynold number and angle of attacks are quite large, 2 to 6 million for Reynold number and -2.5 to 15 for the angle of attack. But only 800 samples are in the training set. Can this small amount of dataset cover the whole data domain?
5) In Figure 3, the lift coefficients are evenly distributed in the whole domain but most of the drag coefficients are distributed at low values. Is this distribution good for the study of this problem? Could this be a reason that the models can not perform well in the drag coeffiecnt?
6) The GitHub link in the paper is broken.

**Documentation:**

The URLs of the data and OpenFOAM are provided, but the link of github code in the paper is broken.

**Ethics:**

There are no ethical issues in the paper.

**Relation To Prior Work:**

The difference between this work and the authors' previous work in ICRL workshop 2022 is not discussed clearly.

**Summary And Contributions:**

This work presents a dataset to study the 2D RANS equations over airfoils with different airfoils, Reynold numbers, and angles of attacks regarding four different tasks.

The authors already address all my concerns. The supplementary material is very detailed. More importantly, the Github is working now and another Github for simulation is provided. This is good for the reproductivity of this work. I would raise my score to 7.

---

> ### Author Response · Authors · 2022-08-15
> **Response to reviewer cACM**
>
> We thank the reviewer for the feedback. We answer the questions below.
>
> Weaknesses:
> 1. The task of optimizing airfoils in a subsonic regime is a classical problem of aerodynamics and often the chosen toy problem. It contains the complexity of RANS equations, the richness of airfoil geometries, and the simplicity of 2D steady-state problems. We added a paragraph called “Design-oriented dataset” in section 3 Dataset Presentation to clarify the motivations of this dataset.
>
> 2. We are very sorry about that, we made a mistake when submitting the main paper by including the unfinished supplementary materials. We submitted a finished version at the supplementary materials deadline and we have now submitted an updated version. This version includes the lacking sections and the additional results for the different models over the different tasks. In particular Table 16 and 17 in the supplementary materials condense all the scores in two tables.
>
> Additional Feedback:
> 1. Thank you for underlying it, we replaced it by “the four fields $u_x$, $u_y$, $p$ and $\nu_t$.
>
> 2. This dataset is a high fidelity version of the precedent dataset that we submitted at the GTRL workshop at ICLR 2022 as it is composed of simulations with finer meshes. The fine meshes allow to limit the numerical diffusion during the resolution and to recover more accurate spatial fields and the trail of airfoils. Moreover, as we use a low-Reynolds approach (i.e. we completely resolve the boundary layers without the help of wall functions), we can accurately recover the force coefficients which was not the case for the previous version. Finally, the turbulence model used here (k-w SST) is different and more accurate than the one chosen for the previous paper (Spalart-Allmaras). A comparison of these two turbulence models is given in section K Simulation Validation of the supplementary materials. Finally, we added a short comment about the differences between AirfRANS and the previous dataset in section 1 Introduction (lines 31 to 34) to clarify it.
>
> 3. Thank you again for pointing it out, Figure 2 has been updated and Figure 1a is given in full size in Figure 5 in the supplementary materials.
>
> 4. Dataset statistics are given in Figure 4 of the supplementary materials. It shows the spanning of the design space (for NACA airfoils and initial conditions) by the dataset. We argue that as the sampling is uniform over each parameter, it is sufficient to have 1000 to correctly span the design space. Moreover, we do not want to give access to a huge quantity of data as one main challenge of this dataset is to encourage the Physics & ML community to build models that are leveraging structures of the problem to better perform on a small amount of data.
>
> 5. For low values of the angle of attack, the main contribution to airfoil drag is due to viscous friction at the wall. In such regime, the boundary layer is thin and attached, and the resulting viscous drag (and, consequently, the drag coefficient) is weakly dependent on the angle of attack that governs the external pressure gradient along the airfoil chord. At higher values of the angle of attack, strong adverse pressure gradients develop along the airfoil, and the boundary layer thickens and eventually separates, leading to the formation of a recirculation bubble at the airfoil upper surface, characterized by low pressure values. The total drag on the airfoil is now contributed by both viscous friction and so-called pressure drag, due to the pressure force component along the flow direction. At such conditions, the drag coefficient increases quickly with the angle of attack, reaching a peak around stall conditions. This should not be a problem for surrogate models if models have seen such patterns in the training (it could be especially challenging in the angle of attack extrapolation regime).
>
> 6. The GitHub repository to reproduce the training and results is now up. Moreover, we propose another GitHub repository to run new simulations.

---

### Official Review · Reviewer_aPMk · 2022-07-28
**Very intersting dataset, but with some shortcomings**

**Rating:** 7
**Confidence:** 3

**Strengths:**

1. the paper presents dataset for a very concrete use case and relevant use case
2. the authors uses openFOAM, which is public available and widely used simulation software
3 the data cover various scenarios
4. the authors provide solid baselines
5. accurate description of the underly equation, the generation of the mesh and preparation
6. analysis of the performance of the models
7. the paper is well written


**Weaknesses:**

While I really like the application and the provided dataset and baseline, I see currently the following limitations:

1. the code of the baselines is not currently available
2. it is not clear it the authors will provide the set up of the openFOAM simulation in order to extend the dataset; this is a critical aspect
3. the dataset itself is limited in number of samples. 1000 sample for 3 and 4 dimensional parameters seems rather limited
4. the maintenance of the data is not clear
5. the authors describe the Langley NASA data, but they seems not to be used neither to validate the basic simulation, neither the trained models
6. the authors provides simulation for the RANS model, but do not consider and provide data with not averaged pde, since the authors do not provide the simulation data with these alternative models a better motivated explanation is needed or some sample simulation would be interesting to have;
7. the simulation have been done by testing outside the training interval, would be nice to have also testing inside the range of the training set (but not overlapping)
8. it would be appreciated the discussion on the "derived" variables
9. it is not clear the values used in the sampling. the values seem arbitrary
10. the authors describe that the jacobian of the velocity is used; I am not sure I understand: a) how are they computed and b) how are they used
11. some references in the introduction are missing (first three sentences)
12. The authors should improve the analysis of the low performance on the drag component, currently it is merely stated that the models do not performe well
13. it would be appreciated an analysis of why MLP works so well; is it because the mesh voxels are located mostly in the same locations for all simulations?
14. since the authors used a open source simulation, more effort could be implemented in generate richer data
15. how have the authors validated that the generate simulation are physically realistic?

I am happy to change my evaluation in the previous points

**Additional Feedback:**

The application is very intersting and I hope the paper meets the standards to be published.

**Clarity:**

Yes, the paper is in general well written (a part from missing references in the introduction) and the problem well presented along with the procedure and mesh generation.

I would thought like more detail on the physical equations and alternative to the average model approach.

**Correctness:**

The experiments set up and their execution is sound, but the sample size is probably too small (1000 samples in total). I am aware of the space required is a critical aspect.

**Documentation:**

Unfortunately is not clear the maintenance process, how long the data will be maintained and how and if it will be extended or maintained

the current data consists of two files: the pre-process data and the openFOAM data



**Ethics:**

there is a section on Broader impact and one sentence on potential negative societal impacts

there seems not be additional ethical consideration, since the data is simulated.

**Relation To Prior Work:**

yes

**Summary And Contributions:**

The dataset contains the description and data from a classical aerodynamic problem, the airfoil simulation.

The authors use consolidated procedure to simulate the airfoil using the two of the NACA airfoil family descriptions.

The authors also describe the underlying CFD equation used in simulation, in particular the RANS model and provide details on the variables and parameters.

The authors provide dataset pre-processed to be use for machine learning task and also the original data from the numerical simulation tool.

The authors use the openFOAM simulation tool, a opensource simulator.

EDIT: I increased my score 6 -> 7

---

> ### Author Response · Authors · 2022-08-15
> **Response to Reviewer aPMk**
>
> We thank the reviewer for their warm encouragement and we hope to address their concerns in the answers below.
>
> Weaknesses:
> 1. The GitHub repository to reproduce the results is now up.
>
> 2. Yes, another GitHub repository is now available to run new simulations or to generate a similar dataset.
>
> 3. It is actually a feature of the dataset. The goal of this dataset is to get close to practical conditions. We find that 1000 simulations are already enough. Hence, it is probable that we will not add more simulations to it. However, a GitHub repository is given to run new simulations and reproduce a similar dataset. Through this setup, we would like to encourage the Physics & ML community to build models that take advantage of the symmetries or the structures of the problem to better perform with a small amount of data.
>
> 4. The dataset is available within the server of ISIR at SCAI laboratory / MLIA team at Sorbonne Université. If we change the server, we will make an update in the GitHub repository and the OpenReview page of the paper. Moreover, if the paper is accepted, we would like to get in touch with the team of PyTorch Geometric to make it directly available via the library.
>
> 5. We argue that  we validate our simulations on two cases of the Langley NASA data; we present it in the section K Simulation Validation of the supplementary materials.
>
> 6. At the considered Reynolds numbers, Direct Numerical Simulations (DNS, which solve all of the turbulent scales in the flow) or Large Eddy Simulations (LES, which solve the most energetic scales of turbulence while modeling the remaining ones) are unfeasible, since they both require solving the whole time evolution of the flow over a long integration time (to ensure convergence of the averaged quantities) and on a 3D grid. The number of grid points scales with $\text{Re}^{9/4}$ for DNS and $\text{Re}^{9/5}$ for wall-resolved LES. The corresponding computation cost of a single simulation would correspond to tens of millions of CPU hours, making a single numerical simulation extremely challenging and the generation of a whole dataset is completely unfeasible. Moreover RANS simulations are accurate enough to find back the drag and lift coefficients which are the most important quantities for this problem.
>
> 7. The full and scarce data regimes are designed in this way, they test the interpolation capacity of models whereas the Reynolds and angle of attack extrapolation regimes are designed to test the capacity of models on out-of-distribution initial conditions.
>
> 8. We call “derived” quantities all the quantities that can be computed from the regressed fields. For example, the velocity gradient (a.k.a. jacobian of the velocity) or the wall shear stress, or the force coefficients. You can find the definition of those quantities in section G Force Coefficients of the supplementary materials or in the (ref to aerodynamic book).
>
> 9. In the paper, we motivate the design space of the initial conditions to reproduce the panel of flight conditions encountered in subsonic flight. We stop at a Mach number of 0.3 (Reynolds number of roughly 6 millions) as the incompressible assumption would not be verified anymore and we start at a Reynolds number of 2 millions because it is a correct lower bound of flight velocity (around 60 knots). Concerning the angle of attacks, the lower bound, -5 degrees, is chosen such as cambered airfoils have a lift coefficient of roughly 0 and the upper bound, 15 degrees, is chosen to prevent stall and too many unsteady patterns.
>
> 10. The jacobian of the velocity is computed with the help of the CFD mesh and the vtkGradientFilter available in ParaView and PyVista (https://vtk.org/doc/nightly/html/classvtkGradientFilter.html). It is used to compute the wall shear stress which is itself used to compute force coefficients (see section G Force Coefficients in the supplementary materials).

---

> > ### Author Response · Authors · 2022-08-15
> > **Response to Reviewer aPMk**
> >
> > 11. It is now fixed. Thank you.The drag coefficient is poorly approximated as it is mainly driven (especially at low angles of attack) by surface frictions and hence by the wall shear stresses. The computation of the wall shear stress involves the computation of the jacobian of the velocity at the surface of the airfoil. In Figure 12, we show the predicted boundary layers for three random test airfoils in the full data regime. We also see see that the first point is often largely overestimated leading to a very poor approximation of the velocity gradient, leading itself to a poor approximation of the drag coefficient. We discuss this problem in Section 5 Benchmarking Results of the main paper, lines 266 to 276.
> >
> > 12. The MLP is pretty well performing in the full data regime as it is powerful enough to learn to interpolate between the different examples. However, as seen in Table 16 and 17 of Section M Additional Results of the supplementary materials, the MLP performs poorly in extrapolation tasks. We also would like to underline the fact that we are working with 2D hexahedrons (quadrilateral…) and not regular voxels/pixels as the data is unstructured.
> >
> > 13. A GitHub repository is now available for running new simulations. However, as we would like to stick with practical problems, we think that it is a feature of the dataset to not include a huge quantity of data. We hope that it will encourage the Physics & ML community to train models with an appropriate inductive bias in order to overcome the problem of scarcity in downstream tasks.
> >
> > 14. We compared our results with those of the NASA Turbulence Modeling Research group over NACA 0012 and 4412. Our validation is given in section K Simulation Validation of the supplementary materials.
> >
> > Clarity:
> > The details of the RANS equations are given in Section F Reynolds-Averaged-Navier-Stokes equations of the supplementary materials. Moreover, the discussion of LES and DNS is out of scope of this paper as it will lead to heavy computations for the simulations and transient dynamics.
> >
> > Additional Feedback:
> > We really appreciate your encouragement and enthusiasm!

---

> > > ### Comment · Reviewer_aPMk · 2022-08-24
> > > **Thank you for the answers**
> > >
> > > Dear Authors,
> > >
> > > Thank you for addressing my questions:
> > >
> > > 1. I accept the explanation about the complexity of alternative simulation to RANS, it is a pity that we can not have a simulation for the alternative approaches;
> > >
> > > 2. Thank you for submitting the code, this makes the dataset very solid
> > >
> > > 3. About MLP, indeed in the additional results in section M, it is possible to see that the error for MLP in the extrapolation task is higher than the alternative approaches. I think this could be highlighted in the main text, not sure how
> > >
> > > 4. Related to point 1., in the simulation time, you could report the approx time for the dataset train/test . For example the full dataset required 400h, correct? This could justify the small dataset (1000 samples)
> > >
> > > 5. Related to the question from the Reviewer EXMb, you provide 1000 samples for this problem. What is a good way to understand if ML is good enough? For example I understand that a correlation above 0.9 is a good value and is a good proxy to evaluate the “success” of a surrogate model. The correlation on the drag for example is very low. Is this the next challenge?
> > >
> > > Thank you again for the huge effort and very throughly answers.

---

> > > > ### Author Response · Authors · 2022-08-24
> > > > **Response to Reviewer aPMk**
> > > >
> > > > Thank you for your additional comments.
> > > >
> > > > 3.  As the results for the other regimes are given in Appendix M, we find that it would be too heavy to discuss those results in the main paper.
> > > >
> > > > 4. We added a line in Table 4 to underline the fact that it took roughly 20 days to generate the full dataset
> > > >
> > > > 5. Yes, the accurate prediction of the wall shear stress (without directly regressing it) is a difficult challenge and this is clearly the next challenge in our opinion. We added a paragraph "Metrics hierarchy" in the conclusion to quickly present the importance of each metric from a design oriented perspective (lines 327 to 336).

---

> > > > > ### Comment · Reviewer_aPMk · 2022-08-29
> > > > > **Thank you for the additional information**
> > > > >
> > > > > Dear Authors,
> > > > >
> > > > > thank you for the additional inforamtion.
> > > > >
> > > > > Please review the added text before creating the final version (line 329 "recpvery" ).
> > > > >
> > > > > Thank you

---

### Official Review · Reviewer_RE8m · 2022-07-28
**An interesting manuscript, but more edits would be needed.**

**Rating:** 6
**Confidence:** 3

**Strengths:**

The numerical solution of Navier-Stokes equations for fluid dynamics is one of the most important PDEs in science and engineering. Current numerical solvers can be computationally expensive and might take thousands of CPU hours.  Having an accurate surrogate model becomes critical. The major contribution of this work is to provide a dataset that would allow a deep learning-based surrogate model to become possible. The dataset is created by RANS equations and shared in openly accessible storage space.

**Weaknesses:**

1.	From my understanding, a data paper should tell end users, what “knowledge” would you gain out of the data and what you can use the data for his/her own benefits. The last thing that you would like to see is to simply assemble a bunch of data with some benchmark. From that perspective, the authors and his team are doing a good job providing useful knowledge on top of the data and the benchmarks are interesting as well. However, I think they can do better. One example would be that you can design datasets in various challenging levels: easy, medium, and hard. Test baseline models on different challenging levels can not only give us a sense of how the baseline models would perform, but also give future users an idea of how to choose among different datasets.


2.	One of the major purposes of the dataset is for the surrogate model. From that perspective, it would be very interesting to study the performance of the DL models by using different sizes of the samples as training sets, and how the performance of the surrogate model will be impacted.

3.     How computationally expensive to generate those data? Will you and your team further improve the datasets based on those limitations that you provided?


4.     All the data provided in the current repo are simulations, are you planning to use the DL model for real applications? How realistic  your simulations are by comparing with real data? Knowledge gap there?


5.	The GitHub link is not working, please fix it.


**Additional Feedback:**

N/A.

**Clarity:**

There are some typos and errors that could be improved. The appendix of the manuscript is not complete (missing sections J to P)!

**Correctness:**

In general, the submission seems correct to me, but the authors and the team need to address the weakness before publishing this work.

**Documentation:**

 Please fix the broken link.

**Ethics:**

I have no concerns here.

**Relation To Prior Work:**

Yes. The differences are clearly stated in the manuscript.

**Summary And Contributions:**

Authors and the team present AirfRANS, high-fidelity CFD datasets for the RANS problems. Relevant DL baselines and benchmarks are also provided to demonstrate the effectiveness of the datasets. Particular problems of interest are generalization tasks (big and scarce data regime, Reynolds number, and angle of attack extrapolation).

---

> ### Author Response · Authors · 2022-08-15
> **Response to Reviewer RE8m**
>
> We thank the reviewer for their helpful comments and suggestions. We answer the questions and comments below:
>
> Weaknesses:
> 1. We definitely agree with the remark, we added in Table 5 of the Benchmarking results the scores of the GraphSAGE model to underline the difficulty of each task and we added the scores of all the models for each task in the Additional Results section of the supplementary materials.
>
> 2. We also agree with the remark, that one of the major challenges of such tasks is the rarity of data. This leads to the need of building models with inductive bias that manage to perform well in scarce data regimes and the need for a dataset to compare the performance on different quantities of data. For this purpose, we propose to train our models on two subsets of the training set but test it on the same evaluation set. In the full data regime, we train our models on 800 simulations (actually 720, 80 are kept for validation), in the scarce data regime, we train our models on 200 of those 800 simulations (actually 180, 20 are kept for validation) and we test both on the remaining 200 simulations. Table 5 in the main paper reports the scores on these two tasks for the GraphSAGE model and Tables 16 and 17 in the Appendix provide the scores for the remaining models.
>
> 3. It took three weeks for the generation of the dataset with 16 physical cores of an AMD Ryzen Threadripper 3960X, around 30 minutes per simulation. In the future, we will not run simulations with airfoil coming out of the NACA 4 and 5 series as the meshing procedure is built over their parametrizations. Also, as the goal of this dataset is to get close to practical conditions, we find that 1000 simulations are already enough. Hence, it is probable that we will not add more simulations to it. However, a GitHub repository is given to run new simulations and reproduce a similar dataset. Through this setup, we would like to encourage the Physics & ML community to build models that take advantage of the symmetries or the structures of the problem to better perform with a small amount of data.
>
> 4. It is a very hard and costly problem to create a multitude of airfoil models with sensors (which are often intrusive) to create a dataset of real experiments. We do not think that it is possible to build such a dataset. Moreover, it is not common in real applications of complex dynamical systems to have access to a sufficiently large quantity of experimental data. However, we validated our simulations with real experiments done by the Turbulence Modeling Research center of NASA on NACA 0012 and 4412 and we presented the results in section K Simulation Validation.
>
> 5. The GitHub repository is now up, we also provide another GitHub repository to run new simulations.
>
> Clarity:
>
> We are very sorry for this confusion, we made a mistake during the submission of the main paper and we forgot to get rid of the unfinished appendices. The final version of the appendices has been submitted before the deadline of the supplementary materials and we have now submitted an update of it.

---

### Author Response · Authors · 2022-06-10
**Main paper information**

Hello,
This comment to inform you that we forgot to update the title with the new one chosen for the main paper submission i.e. "AirfRANS: High Fidelity Computational Fluid Dynamics Dataset for Approximating Reynolds-Averaged-Navier–Stokes Solutions".

Also to inform you that the entire (main + supplementary) has been uploaded in the same .pdf for the main paper submission. However, please do not pay attention to the supplementary part as it is not finished yet.

Finally, we forgot to include the "Paper Checklist" section in the main paper, we will include it in the supplementary materials.

Sorry about all of that... The deadlines are a bit tight for us. Would it be possible to re-upload a version of the main paper at the supplementary materials deadline for minor changes to be coherent with the supplementary materials form? (and correctly cut the main paper from the supplementary materials)

Kind regards,
Florent Bonnet

---

### Author Response · Authors · 2022-08-15
**General response for all reviewers**

We sincerely thank all the reviewers for their efforts to review our submission and for the several constructive suggestions to improve our paper. We are pleased to see their interest in the topic presented in this work and that the paper has mainly been considered well written and clear.

Moreover, following their feedback, we provided a revision of our initial submission (merging the main paper and the supplementary material), where modifications are colored in red. In summary,
in the main paper:
-    We added a short comment in the introduction (lines 31 to 34) to clarify the difference between the AirfRANS dataset and its precedent version submitted at the GTRL workshop of ICLR 2022
-    In section 3 Dataset Presentation, we added a paragraph called Design-oriented dataset (lines 82 to 87) to clarify the motivation of the dataset and a short sentence (lines 144 to 145) to justify the relatively small quantity of data available in the dataset. We also updated Figure 2 for more readability.
-    In section 5 Benchmarking Results, we updated all the tables with new scores of training that are coherent with new experiments conducted on the four tasks and scores given in the GitHub repository. There is no significant difference between them and the ones presented in the first submission. Moreover, we added Table 5 where we give the scores of the GraphSAGE model for the four different tasks. A paragraph “Difficulties of the different tasks.” (lines 277 to 290) has also been added to discuss the results of Table 5.
-   In section 6 Conclusion, a short comment has been added (lines 323 to 327) to propose neural operators acting on unstructured data as potential candidates for surrogate models.
-   We correctly added the Paper Checklist (section 7) in the main paper instead of the supplementary materials.

In the supplementary materials:
-   The section C Reproducibility Statement has been updated (lines 625 to 633) with new links to GitHub repositories for reproducing experiments and running new simulations
-   In the section M Additional Results has been updated with the scores of all the models for the four different tasks with all the plots associated with them. A paragraph Summary (lines 1011 to 1021) has been updated to briefly comment on those new scores.

In addition to that, we tried to correct grammatical errors and slightly change certain sentences when we thought the formulation was too heavy.

For the concerns about the maintenance of the dataset, we would like to emphasize that the dataset is available within the server of ISIR at SCAI laboratory / MLIA team at Sorbonne Université. If we change the server, we will make an update in the GitHub repository and the OpenReview page of the paper. Moreover, if the paper is accepted, we would like to get in touch with the team of PyTorch Geometric to make it directly available via the library. Also, for the moment, we do not want to add more simulations to this dataset as the reduced quantity of data is a feature of it instead of a limitation.

Finally, we would like to sincerely apologize for uploading the Appendix sections in the first submission of the main paper. This has been a source of confusion.

---

### Meta-Review · Area_Chair_MLxp · 2022-09-14

**Recommendation:** Accept
**Confidence:** 3

**Metareview:**

This paper provides high-fidelity datasets for computational fluid dynamics, based on Reynolds-Averaged-Navier–Stokes simulations over airfoils. While it is possible for anyone to generate such a dataset, doing so is expensive and hence, having a reference dataset can be helpful for lowering the bar for entry for researchers. The paper also provides graph neural net baselines on 4 different dataset regimes, which can help in diverse benchmarking for future works. The dataset and code is open-sourced for further use. There were some important concerns raised by the reviewers regarding the missing details on dataset construction and use, which were appropriately addressed by the authors.

---

### Decision · Program_Chairs · 2022-09-16

Accept